# Genetic Iron Overload Hampers Development of Cutaneous Leishmaniasis in Mice

**DOI:** 10.3390/ijms24021669

**Published:** 2023-01-14

**Authors:** Edouard Charlebois, Yupeng Li, Victoria Wagner, Kostas Pantopoulos, Martin Olivier

**Affiliations:** 1Lady Davis Institute for Medical Research, Jewish General Hospital, Montreal, QC H3T 1E2, Canada; 2Department of Medicine, McGill University, Montreal, QC H4A 3J1, Canada; 3Faculty of Veterinary Medicine, Université de Montréal, Montreal, QC J2S 2M2, Canada; 4Research Institute of the McGill University Health Centre, Montreal, QC H4A 3J1, Canada

**Keywords:** leishmaniasis, iron, hemojuvelin, hepcidin, hemochromatosis, macrophages

## Abstract

The survival, growth, and virulence of *Leishmania* spp., a group of protozoan parasites, depends on the proper access and regulation of iron. Macrophages, *Leishmania’s* host cell, may divert iron traffic by reducing uptake or by increasing the efflux of iron via the exporter ferroportin. This parasite has adapted by inhibiting the synthesis and inducing the degradation of ferroportin. To study the role of iron in leishmaniasis, we employed *Hjv*^−/−^ mice, a model of hemochromatosis. The disruption of hemojuvelin (Hjv) abrogates the expression of the iron hormone hepcidin. This allows unrestricted iron entry into the plasma from ferroportin-expressing intestinal epithelial cells and tissue macrophages, resulting in systemic iron overload. Mice were injected with *Leishmania major* in hind footpads or intraperitoneally. Compared with wild-type controls, *Hjv*^−/−^ mice displayed transient delayed growth of *L. major* in hind footpads, with a significant difference in parasite burden 4 weeks post-infection. Following acute intraperitoneal exposure to *L. major*, *Hjv*^−/−^ peritoneal cells manifested increased expression of inflammatory cytokines and chemokines (*Il1b*, *Tnfa*, *Cxcl2*, and *Ccl2)*. In response to infection with *L. infantum*, the causative agent of visceral leishmaniasis, *Hjv*^−/−^ and control mice developed similar liver and splenic parasite burden despite vastly different tissue iron content and ferroportin expression. Thus, genetic iron overload due to hemojuvelin deficiency appears to mitigate the early development of only cutaneous leishmaniasis.

## 1. Introduction

*Leishmania* spp. are sandfly-transmitted trypanosomatid protozoan parasites endemic to tropical and sub-tropical regions including the Mediterranean area, which rely on iron for growth and differentiation [1,2,3]. The pathology of this disease can range from self-healing cutaneous lesions to lethal visceralizing disease, depending on *Leishmania* species and strain. Infection with *Leishmania major* results in cutaneous leishmaniasis, while *Leishmania infantum* will spread into organs. The propagation of this disease is increasing due to environmental changes and socio-economic conflicts, with an estimated 0.7–1 million new cases annually [4]. In humans, parasites infect macrophages or neutrophils, later to be engulfed by macrophages infecting them in a trojan horse mechanism [5,6]. Macrophages are equipped with multiple mechanisms to combat this intracellular infection, including diverting the iron flux to starve invaders [7,8]. Iron efflux is mediated by the sole cellular iron exporter ferroportin (gene name *Slc40a1*). *Leishmania* spp. have adapted to this challenge by inhibiting *Slc40a1* mRNA translation and by promoting ferroportin degradation [9,10].

Iron is an essential micronutrient for practically all living organisms and a central component of heme groups, iron–sulfur clusters, and the key enzymes involved in mitochondrial respiration and DNA synthesis. Heme is of particular importance, as *Leishmania* spp. are heme-auxotrophs [11]. Furthermore, parasite differentiation from promastigote to amastigote is dependent on iron, whose availability drastically differs between the vector and the host [12]. Susceptibility to *Leishmania* infection has been associated with Nramp1 (gene name *Slc11a1*), an iron transporter on the parasitophorous vacuole membrane, suggesting an important role of host iron metabolism for parasite growth [13,14,15]. In experiments with rodent models, pharmacological treatments with the iron chelator desferrioxamine had either no effect [16] or suppressed [17,18,19] intra-macrophagic growth, depending on parasite species. The pretreatment of mice with desferrioxamine only seemed to cause a slight delay in the growth of cutaneous lesions [20]. Comparably, the dietary iron restriction had very little impact on the proliferation of the visceral disease-causing species *L. infantum* [21].

Conversely, iron loading may cause parasite killing due to the formation of reactive oxygen species (ROS), which may overwhelm parasite defenses. Thus, iron administration in murine models limited the growth of both cutaneous [20,22,23] and visceral [21] disease-causing strains of *Leishmania.* Iron loading may also play a role in host immunity, which can modulate NF-κB signaling in macrophages [24,25]. This pathway would then induce the differentiation of IFNγ-producing CD4^+^ T cells crucial for parasite restriction [26].

In general, iron is an important regulator of immune responses. Excess iron may either impair [27,28] or induce [29] pro-inflammatory cytokine production in macrophages. In cell culture experiments, iron supplementation favored Th2 activation and antagonized IFNγ responses [30]. Yet, iron was also shown to directly drive T-helper-cell pathogenicity through its interactions with the iron chaperone poly(rC)-binding protein 1 [31]. Nitric oxide production, which is necessary for intracellular pathogen killing, greatly affects cellular iron homeostasis and promotes iron accumulation [28,32]. Iron negatively regulates the transcription of the inducible nitric oxide synthase, providing a feedback mechanism [33].

Systemic iron metabolism is primarily controlled by hepcidin (gene name *Hamp*), a hepatocyte-derived peptide hormone [34]. Hepcidin binds to ferroportin, occludes its opening [35], and targets it for lysosomal degradation [36]. This inhibits iron import to the bloodstream from absorptive intestinal enterocytes and erythrophagocytic tissue macrophages. Hepcidin expression is predominantly regulated at the transcriptional level in response to iron fluctuations through BMP/SMAD signaling, or in response to inflammatory cues via JAK/STAT signaling, initiated largely by IL-6 [37].

Hemojuvelin (Hjv) is a bone morphogenetic protein (BMP) co-receptor that enhances BMP/SMAD signaling to hepcidin on hepatocytes. The disruption of the *HJV* gene causes juvenile hereditary hemochromatosis in humans [38], and a similar phenotype is observed in *Hjv*^−/−^ mice [39]. Hereditary hemochromatosis comprises a group of genetically heterogenous disorders of systemic iron overload caused by hepcidin suppression [40]. Paradoxically, tissue macrophages are unable to retain iron [41] due to the unrestricted expression of ferroportin on the cell surface [39,42]; this results in lower splenic iron content. Macrophage iron deficiency, combined with concurrent systemic iron overload, provides a unique environment to study leishmaniasis. Herein, we sought to explore whether the hemochromatosis phenotype favors resistance to *Leishmania* as has been previously reported for other intra-macrophage pathogens [43,44].

## 2. Results

### 2.1. Growth of Leishmania Major Is Transiently Delayed in Hjv^−/−^ Mice during Early Infection

We first sought to assess the susceptibility of genetically iron-overloaded mice to cutaneous leishmaniasis. We injected wild-type (*Hjv^+/+^*) control and *Hjv*^−/−^ mice with *L. major* in hind footpads. Footpad swelling was followed over a period of 8 weeks (Figure 1A). Statistical differences in swelling between both groups were observed for all but the final week (Figure 1A), indicating that *Hjv*^−/−^ mice are at least less sensitive to infection with *L. major*. A limiting dilution assay revealed that parasite load was significantly reduced in the footpads of *Hjv*^−/−^ mice 4 weeks post-infection but not at 7 weeks (Figure 1B,C). These data suggest that parasite load recovers before swelling.

Popliteal lymph nodes, the draining lymph nodes that are accessible to the parasite during footpad infections, were collected and evaluated for cytokine and chemokine gene expression. In both mouse models, cytokine gene expression appeared to increase over time. It was significantly higher than in mesenteric lymph nodes (MLN) used as uninfected control tissues, except for *Il6* expression, which remained at the baseline (Figure 2A–D). Chemokine expression remained largely unaffected in lymph nodes, with only a slight increase in *Ccl2* expression at 7 weeks post-infection (Figure 2E–H). No genotype-specific differences were observed, suggesting that cytokine and chemokine expression in the draining lymph node is not the cause of the delayed parasite growth observed in footpads.

### 2.2. Cytokine Acute Response to Leishmania Major Is Altered in Hjv^−/−^ Mice

Considering that lymph node cytokine gene expression was similar across both genotypes, we hypothesized that early parasite establishment within the host could be a main factor in the observed relative resistance of *Hjv*^−/−^ mice to *L. major*. To study this, we injected the mice intraperitoneally with *L. major* and collected the serum, liver, and peritoneal lavage 6 h post-infection. The iron parameters in the circulation were measured to better understand the systemic effects of *Leishmania* infection on iron distribution. The bacterial endotoxin lipopolysaccharide (LPS) was used as a control for inflammation. Serum iron from the mice infected with *L. major* was unaffected compared with the drop observed by LPS in the wild-type animals (Figure 3A), which is known as a hypoferremic response to inflammation [8]. Transferrin saturation, another marker of circulating iron levels, displayed a similar trend (Figure 3B). Characteristically, iron levels were largely unchanged throughout all inflammatory treatments in *Hjv*^−/−^ mice, and transferrin saturation was at maximal levels, with only a slight reduction observed after LPS treatment (Figure 3A,B), as previously described [45]. *Leishmania* infection did not alter total iron binding capacity (TIBC) in either wild-type or *Hjv*^−/−^ mice (Figure 3C).

The liver plays an important role in the immune response to pathogens by being the major producer of hepcidin to control systemic iron traffic. Invading extracellular bacterial pathogens will typically activate the expression of *Hamp* mRNA due to IL-6-driven inflammation [46]. The ensuing hypoferremic response is thought to inhibit the growth of pathogens by depriving them of iron. Yet, for intracellular pathogens, this would be deleterious, as iron levels would consequently increase within infected cells. In fact, host macrophages are known to upregulate ferroportin in order to reduce their iron content in response to intracellular pathogens such as *S. typhimurium* [47]. However, this has not been observed in the context of leishmaniasis, as parasites will suppress ferroportin synthesis and will also increase hepcidin expression over the course of infection [9,10]. Interestingly, *Hamp* mRNA levels were measured and were increased in response to LPS, but not to *L. major* in wild-type mice, in this experimental time frame (Figure 3D). *Hamp* expression was significantly reduced in *Hjv*^−/−^ mice (Figure 3D), as expected. The liver expression of the inflammatory cytokine-encoding *Il6*, *Il1b*, and *Tnfa* mRNAs was unresponsive to *L. major* infection compared with endotoxin exposure (Figure 3E–G), whereas *Ifng* expression was only induced by the parasite in wild-type control mice (Figure 3H). Levels of *Il6*, *Il1b*, and *Tnfa* mRNAs did not differ in the livers of LPS-treated wild-type and *Hjv*^−/−^ mice, as previously reported [45].

We then assessed cytokine gene expression in the peritoneal lavage. LPS-treated animals responded as anticipated with increased expression of all tested cytokine and chemokine genes, except for *Ifng* (Figure 4). A similar trend was observed in *L. major*-infected animals with the additional lack of response in *Il6* (Figure 4A). Interestingly, there was an overall trend toward the upregulation of cytokine and chemokine gene expression in *Hjv*^−/−^ mice (Figure 4). *L. major* infection produced a less pronounced effect overall, without any remarkable differences between genotypes in the expression of *Il6*, *Ifng*, *Ccl3*, and *Ccl4* mRNAs (Figure 4). However, it should be noted that IL-6 and CCL2 protein levels were elevated in knockout mice in peritoneal lavage supernatants when measured with a multiplex assay (Appendix A).

Analysis of the cell suspension via cytospin centrifugation revealed that the peritoneal lavage 6 h post-infection mostly consisted of neutrophils and macrophages (Figure 5A). We utilized these cell populations to assess the rate of infection and parasite load per cell immediately following separation (Figure 5B,C) or after allowing the collected cells to grow in culture for 24, 48, and 72 h (Figure 5D,E). Under these conditions, the cultured cells were primarily macrophages, which clear neutrophils through phagocytosis. No genotype-specific differences were observed in the infection rate or the cellular parasite burden at any timepoint measured, suggesting that *Hjv*^−/−^ cells do not exhibit altered phagocytosis, consistent with another report [48]. In addition, there was no increased intracellular parasite killing at these timepoints.

### 2.3. Genetic Iron Overload Does Not Impact Visceral Disease Progression by Leishmania Infantum despite Induction of the Iron Exporter Ferroportin

*L. infantum* causes visceral leishmaniasis affecting primarily the liver and the spleen. To assess the role of genetic iron overload in the progression of visceral disease, wild-type and *Hjv*^−/−^ mice were infected with *L. infantum* and sacrificed 1, 2, or 3 weeks post-infection. Organs were collected, weighed, and used for the limiting dilution assay (Figure 6A–D). No differences in organ weight were observed between the genotypes (Figure 6A,C), and the organ parasite burden was similar at the measured timepoints (Figure 6B,D). The splenic iron content was characteristically low in knockouts as previously reported [39,42] (Figure 6E), and ferroportin protein expression was significantly elevated (Figure 6F). Nevertheless, parasite growth and disease progression were largely unaffected.

## 3. Discussion

Hemochromatosis resulting from the disruption of *Hjv* produces a unique iron environment where enterocytes and macrophages cannot retain iron and release it to the bloodstream. Thus, excessive amounts of metal accumulate in the plasma and are eventually taken up by tissue parenchymal cells. In the present work, we sought to understand how this iron environment would affect the growth of *L. major* and *L. infantum*, the parasites causing cutaneous and visceral leishmaniasis, respectively. The delayed swelling of *L. major*-infected *Hjv*^−/−^ mice (Figure 1) closely resembles the results with desferrioxamine-pretreated mice, showing a several-week delay in parasite growth [20]. Consequently, it is likely that the microenvironment within *Hjv*^−/−^ murine macrophages mimics that of the macrophages from desferrioxamine-pretreated mice and is characterized by relative iron deficiency. Furthermore, inappropriately low circulating hepcidin results in excessive ferroportin expression at the cell surface of macrophages, as observed in splenic extracts in Figure 6F. Despite having decreased iron stores, the macrophages from *Hjv*^−/−^ mice exhibited physiological clearance of senescent red blood cells (erythrophagocytosis), which implies high iron turnover. In another model of iron overload, involving iron dextran injection into wild-type mice, macrophages became extremely iron-loaded. Under these conditions, *L. major* growth was inhibited due to increased parasite killing [20], which was possibly enhanced by ROS-mediated shift in immunity toward a T-helper type 1 (Th1) response [23]. Thus, it is difficult to directly compare data from the genetic and pharmacological models of iron overload since they significantly differ in tissue iron distribution.

Notably, the genetic background of mice may also affect immune responses and leishmaniasis progression. Herein, we used C57BL/6 mice, which tend to favor a Th1 response resulting in resistance to persistent infection; on the other hand, BALB/c mice favor Th2 responses leading to susceptibility [49,50]. These effects are also dependent on *Leishmania* spp., which modulate macrophage immunity, adding another level of complexity. For instance, in a recent study, infection with *L. panamensis* induced a potent activation of classical M1 macrophages in C57BL/6 mice but only an intermediate response in BALB/c mice [51]. Hence, it would be of interest to study the effects of genetic iron overload using different mouse and *Leishmania* spp. strains.

We did not observe any induction of the *Ifng* gene through either LPS treatment or *L. major* infection in the macrophages and neutrophils collected from the peritoneal lavage of both wild-type and *Hjv*^−/−^ mice 6 h post-treatment, suggesting that these cells are not the major producers of IFNγ (Figure 4D). This result somewhat contradicts previous data, where a marked reduction in IFNγ levels was observed 6 h post-*E. coli* infection in sera as well as cultured thioglycolate-elicited peritoneal macrophages from *Hjv*^−/−^ vs. wild-type mice [48]. The disparity between these findings may lie in the different analyzed sections (peritoneal cells vs. sera) as well as the mouse model used. Interestingly, in our experiments, liver *Ifng* expression was significantly diminished 6 h post-infection in *Hjv*^−/−^ mice (Figure 3H), suggesting that perhaps tissue-resident Th1 or NK T cells could be an important source of IFNγ, which then enters the circulation. In fact, Th1 cells [52] and NK T cells [53] are considered the primary producers of IFNγ. Whether macrophages and neutrophils can produce IFNγ remains contentious, and in vitro experiments with primary cells may not always be physiologically relevant [54]. Taken together, our data do not provide any evidence that macrophages or neutrophils can produce IFNγ, at least in our experimental setting. However, we cannot rule out the possibility that the circulating IFNγ produced elsewhere is an important determinant for the establishment of *Leishmania* infection in footpads.

We noted a marked increase in the expression of many cytokine and chemokine genes in the peritoneal cells of *Hjv*^−/−^ mice (Figure 4). Cytokines play a differential role in infection, depending on mouse strain as well as parasite species. *Tnfa* expression has been linked to the protection against cutaneous leishmaniasis during the early stages of infection [55], whereas it can lead to immunopathology when it persists later in the disease [56]. This makes it a strong candidate for the delayed parasite growth observed during the early weeks of infection in knockout animals. A multiplex assay was performed to measure GM-CSF, IFNγ, IL-1β, IL-2, IL-4, IL-6, IL-10, IL-12p70, CCL2, and TNFα. Only IL-6 and CCL2 were consistently expressed in lavage supernatants of infected mice, and their levels were significantly higher in *Hjv*^−/−^ mice (Appendix A). IL-6 has been reported to play a dual role in leishmaniasis. It may downregulate macrophage leishmanicidal effects [57], while it may also induce the growth of IL-10+ CD4+ T cells [58]. Considering the different species of *Leishmania* studied in these analyses, it is difficult to specify the exact role of IL-6 in the development of cutaneous lesions from *L. major* infection in our study. Given its effects on macrophages, it is unlikely that IL-6 accounts for the observed delayed growth of cutaneous lesions in *L. major*-infected *Hjv*^−/−^ mice. CCL2 is an important chemoattractant for monocytes. Yet, it may be involved in Th2 polarization, as *Ccl2*^−/−^ mice are resistant to *L. major* and do not have abnormal naïve T-cell migration [59]. This is in contrast to its cognate receptor CCR2, which plays a crucial role in the protection against cutaneous leishmaniasis [29]. Thus, we rationalize that CCL2 would be protective during early *L. major* infection by recruiting monocytes but may later be detrimental. Taken together, *Hjv*^−/−^ mice may be protected by the production of TNFα, even though this cytokine could not be directly measured in lavage supernatants, and by the enhanced expression of CCL2.

Remarkably, *Hjv*^−/−^ mice were not protected from the visceral disease resulting from *L. infantum* infection despite having greatly increased ferroportin levels in the spleen, as well as reduced splenic iron content (Figure 6E,F). These results corroborate earlier data showing that dietary iron restriction did not affect parasite load in the mouse liver and spleen, 60 days post-*L. infantum* infection [21]. The stark difference in these two models is that dietary iron deficiency does not induce, and rather post-transcriptionally suppresses, tissue ferroportin [60], the expression of which is reported to be protective [9]. Interestingly, pharmacological iron deficiency induced by a two-week pretreatment of BALB/c mice with desferrioxamine [61] led to a significant decrease in the splenic parasite load 6 weeks post-*L. chagasi* infection. Taken together, these results suggest that the severity and timeframe of iron deficiency may be important for parasite replication, particularly at the later stages of infection. Nevertheless, the iron restriction does not appear an optimal strategy for the control of visceral leishmaniasis.

Leishmaniasis is a complicated disease considering that iron supplementation seems to protect the host against the parasite [20,21,22,23], while iron deprivation appears to have little to no effect [20,21], contrary to most other pathogens [7,8]. The hemochromatosis phenotype of systemic iron overload with macrophage iron deficiency and high iron turnover did not further increase vulnerability to *Leishmania* infection as has been reported with several other bacteria, fungi, and even viruses [62]. Previous publications showed that *Leishmania* spp. express many different receptors and transporters for iron [63,64,65,66,67]; nevertheless, it remains to be clarified which form of iron is the most important for amastigote development and replication. Herein, we provide evidence that transient iron pools in macrophages are sufficient for parasite replication, even with enhanced iron export, resulting in net iron deficiency. Our work cannot differentiate between the systemic effects of *Hjv* deficiency on iron metabolism and possible local immunological effects in macrophages. This remains to be addressed in future studies.

## 4. Materials and Methods

### 4.1. Animals and Ethics

Mouse experiments were performed in the McGill University Health Centre research institute in containment level 2 housing facilities. Wild-type C57BL/6J and isogenic *Hjv*^−/−^ mice [68] were housed under pathogen-free conditions in macrolone cages (up to 5 mice/cage, 12:12 h light–dark cycle: 7 a.m.–7 p.m.; 22 ± 1 °C, 60 ± 5% humidity). At the endpoints, the animals were sacrificed through CO_2_ inhalation and cervical dislocation. Isoflurane was used for anesthesia prior to euthanasia to alleviate suffering. All the mice used in the experiments were male.

### 4.2. Parasite Culture

*L. major* (strain MHOM/SN/74/Seidman) promastigotes were generously supplied by Robert McMaster (University of British Columbia, Vancouver, BC, Canada). All parasites were cultured at 25 °C, 5% CO_2_ in Schneider’s Drosophila Medium (SDM) supplemented with 10% heat-inactivated fetal bovine serum (FBS, Wisent, St-Bruno, QC, Canada), and 5 mg/mL hemin and passaged every 3 to 4 days. The cultures of promastigotes growing at the logarithmic phase (day 3–4 post-passage) were passaged biweekly and were grown to the stationary phase (day 6–8 post-passage) before being used in infections for all the experiments [69].

### 4.3. Footpad Infections

Groups of ten 6-week-old mice per genotype were each injected with 5 × 10^6^
*L. major* promastigotes into one hind footpad. Five mice per genotype were sacrificed at four and seven weeks, and footpads were used to measure the parasite burden with a limiting dilution assay. Popliteal lymph nodes were also collected, snap-frozen in liquid nitrogen, and later analyzed using qPCR. A group of 5 mice per genotype was kept, and footpads were measured at 8 weeks post-infection. The mice in each group were housed in the same cage for the duration of the experiment. The uninfected footpad was used as the negative control for measurement purposes. Lesion development was monitored weekly by measuring the difference in footpad thickness between the infected and uninfected footpad, measured using digital calipers. The experiments were repeated up to three times.

### 4.4. Limiting Dilution Assay

Footpads were sterilized with ethanol, excised, and washed with phosphate-buffered saline (PBS). Next, tissue was disrupted manually using a glass tissue homogenizer in sterile PBS under a BSL2 tissue culture hood. Briefly, 50 mL total volume of footpad homogenate was recovered, and 100 µL of each sample was added to 96-well plates (Sarstedt, Germany) containing 100 µL complete SDM per well, in duplicate. A minimum of 24 twofold serial dilutions were performed for each sample. The plates were kept at 25 °C until microscopic examination 10 days later, when the highest dilutions at which the promastigotes were observed were recorded.

### 4.5. Acute Intraperitoneal Infections

Groups of 3 mice each were injected intraperitoneally with PBS, 1 μg/g LPS (serotype 055:B5; Sigma-Aldrich, St. Louis, MI, USA), or 10^8^
*L. major* promastigotes before being sacrificed 6 h later. The mice in each group were housed in the same cage for the duration of the experiment. In total, 9 mice were analyzed per genotype per treatment group. The mice were lavaged with 5 mL of ice-cold endotoxin-free PBS at endpoints. The number of live cells present in the lavages was counted using a hemocytometer. The cells were prepared for microscopy using a cytospin 4 cytocentrifuge (Thermo Scientific, Waltham, MA, USA). The cells were fixed and stained using a Differential Quik (Diff-Quik) Stain Kit (Ral Diagnostics, Martillac, France). The percentage of cell types found in the lavage was counted. Next, the percentage of cells infected and the number of *Leishmania* amastigotes found within the cells were counted. Briefly, 150 μL of the lavages were plated in 4-well chamber slides (Corning, USA) and incubated at 37 °C for 1 h before the addition of Dulbecco’s modified eagle medium (Wisent, St-Bruno, QC, Canada), with 10% FBS and 1% penicillin–streptomycin-glutamine. The cells were kept at 37 °C in 5% CO_2_. Then, 24, 48, and 72 h post-plating, the media were removed, and the cells were then air-dried, fixed, and stained using the Diff-Quik Stain Kit. The percentage of infected cells and the number of Leishmania amastigotes found within the cells were counted. From the total 300 cells counted from each slide, the percentage was calculated, and the number of amastigotes found in individual cells was counted as well. The total lavage was centrifuged at 1500 rpm for 10 min to separate cells and supernatant. These fractions were kept at −80 °C until processing.

### 4.6. Visceral Leishmaniasis Infection

Briefly, 8-week-old mice (n = 5–10) were infected intraperitoneally with 10^8^
*L. infantum* promastigotes (MHOM/MA/67/ITMAP-263). The control mice were injected with PBS. The mice were weighed and sacrificed 1, 2, and 3 weeks post-infection. The whole livers and spleens were collected, weighed, and snap-frozen in liquid nitrogen. A piece of the liver and spleen was weighed and directly used without freezing for the limiting dilution assay. The spleens were later used for Western blotting and iron quantification.

### 4.7. Serum Biochemistry

Blood was collected via cardiac puncture. The serum was prepared by using micro Z-gel tubes with a clotting activator (Sarstedt) and was kept frozen at −20 °C until analysis. Serum iron and the total iron binding capacity (TIBC) were determined at the Biochemistry Department of the Montreal Jewish General Hospital using a Roche Hitachi 917 Chemistry Analyzer. Transferrin saturation was calculated from the ratio of serum iron and TIBC.

### 4.8. Quantitative Real-Time PCR (qPCR)

RNA was extracted from organs and cells by using an RNeasy kit (Qiagen). cDNA was synthesized from 1 μg RNA by using the OneScript^®^ Plus cDNA Synthesis Kit (Applied Biological Materials Inc., Richmond, BC, Canada). Gene-specific primer pairs (Appendix A) were validated through the dissociation curve analysis and demonstrated amplification efficiency between 90% and 110 %. SYBR Green (Bioline) and the primers were used to amplify the products under the following cycling conditions: initial denaturation of 95 °C 10 min, 40 cycles of 95 °C 5 s, 58 °C 30 s, 72 °C 10 s, and a final cycle melt analysis between 58 and 95 °C. The relative mRNA expression was calculated using the 2^−ΔΔCt^ method [70]. Data were normalized to murine ribosomal protein L19 (*Rpl19*). Data are reported as fold increases compared with the samples from wild-type mice.

### 4.9. Multiplex Cytokine/Chemokine Quantification Assay

Briefly, 100 μL of the lavage supernatant was analyzed using a Mouse Cytokine Proinflammatory Focused 10-Plex Discovery Assay^®^ Array (MDF10) (Eve Technologies, Calgary, AB, Canada). These include GM-CSF, IFNγ, IL-1β, IL-2, IL-4, IL-6, IL-10, IL-12p70, CCL2, and TNFα. The multiplex laser bead technology utilizes the antibodies that are coupled to color-coded polystyrene beads, where lasers activate the fluorescent dye and excite the fluorescent conjugate, which is then quantified for the concentration of the target analyte.

### 4.10. Western Blotting

Spleens were washed with ice-cold PBS and dissected into pieces. Aliquots were snap-frozen in liquid nitrogen and stored at −80 °C. Protein lysates were obtained as described [42]. The lysates containing 40 μg of proteins were analyzed using SDS–PAGE on 9–13% gels, and the proteins were transferred onto nitrocellulose membranes (BioRad). The blots were blocked in non-fat milk diluted in tris-buffered saline (TBS) containing 0.1% (*v*/*v*) Tween-20 (TBS-T) and probed overnight with antibodies against ferroportin [71] (1:1000 diluted monoclonal rat anti-mouse 1C7, kindly provided by Amgen Inc), β-actin (1:2000 diluted; Sigma). Following a 3× wash with TBS-T, the membranes were incubated with peroxidase-coupled secondary antibodies for 1.5 h. Immunoreactive bands were detected via enhanced chemiluminescence with a Western Lightning ECL Kit (Perkin Elmer, Waltham, MA, USA). Blot images were quantified using ImageJ software (version 1.53t).

### 4.11. Tissue Iron Quantification

The splenic iron content (SIC) was quantified by using the ferrozine assay [72].

### 4.12. Statistics

Statistical analysis was performed by using the Prism GraphPad software (version 9.1.0). The lognormally distributed data including qPCR results were first log-transformed before analysis with an unpaired Student’s *t*-test. The normally distributed data were analyzed with an unpaired Student’s *t*-test. Comparisons within the same genotype are denoted by a or b in figures. A probability value of *p* < 0.05 was considered statistically significant.

## Figures and Tables

**Figure 1 ijms-24-01669-f001:**
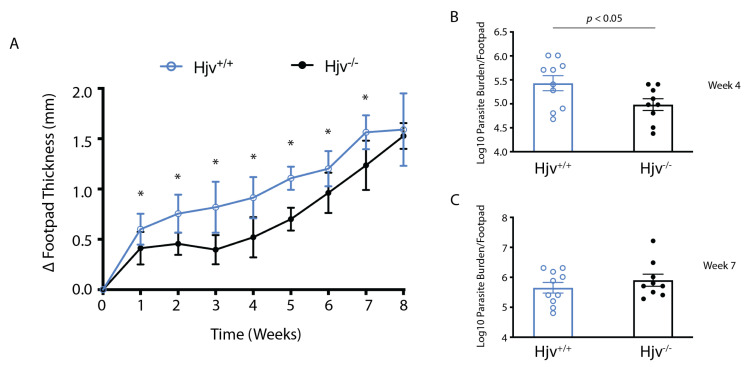
*Hjv*^−/−^ mice exhibit relative resistance to *L. major* footpad infection. *Hjv^+/+^* and *Hjv*^−/−^ mice (n = 5–24 per group) were injected in hind footpads with 5 × 10^6^
*L. major* parasites: (**A**) footpads were measured weekly over 8 weeks, and thicknesses of uninfected versus infected footpads were compared. Footpads were collected at endpoints and used to perform a limiting dilution assay of parasite growth at 4 weeks (**B**), or 7 weeks (**C**) post-infection. Time course data are presented as mean ± SD, while log_10_ number of parasites per footpad are presented as mean ± SEM. * indicates *p* < 0.05.

**Figure 2 ijms-24-01669-f002:**
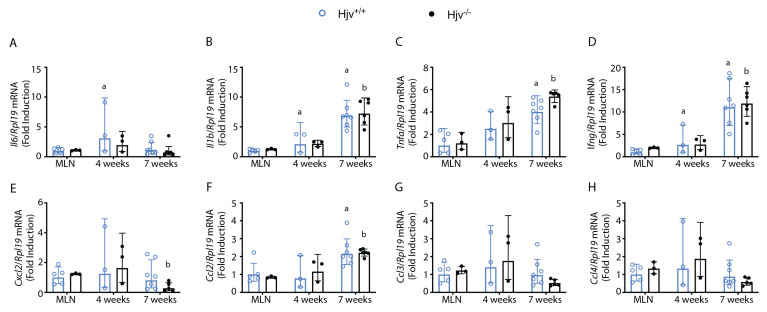
Popliteal lymph node cytokine expression following footpad infection. Popliteal lymph nodes were collected from *Hjv*^+/+^ and *Hjv*^−/−^ mice infected with 5 × 10^6^
*L. major* in hind footpads 4 or 7 weeks post-infection. Mesenteric lymph nodes (MLN) were also collected as uninfected control organs. RNA was extracted, reverse-transcribed, and used for qPCR analysis of (**A**) *Il6*, (**B**) *Il1b*, (**C**) *Tnfa*, (**D**) *Ifng*, (**E**) *Cxcl2*, (**F**) *Ccl2*, (**G**) *Ccl3*, and (**H**) *Ccl4* mRNAs. Data are presented as geometric mean ± geometrical SD. Statistical differences compared with MLN from *Hjv*^+/+^ or *Hjv*^−/−^ mice are indicated by a or b, respectively.

**Figure 3 ijms-24-01669-f003:**
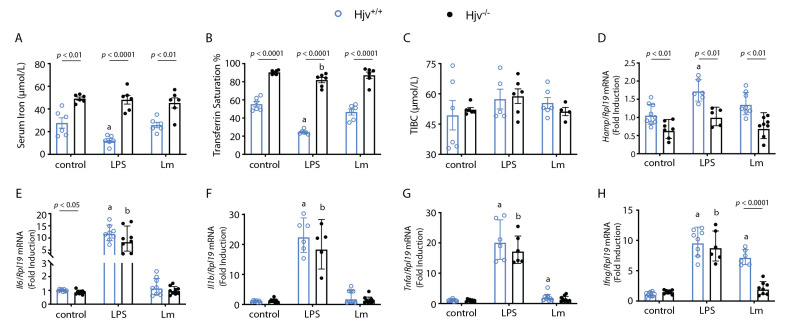
*Leishmania*’s major acute infection does not trigger a hypoferremic response. *Hjv^+/+^* and *Hjv*^−/−^ mice were injected intraperitoneally with either phosphate-buffered saline (control), the endotoxin LPS, or 10^8^
*L. major* stationary phase parasites (Lm). Then, 6 h post-infection, blood was collected via cardiac puncture. Serum was separated from whole blood and used for analysis of (**A**) serum iron, (**B**) transferrin saturation, and (**C**) total iron binding capacity (TIBC). Liver samples were collected; RNA was extracted and used for analysis of (**D**) *Hamp,* (**E**) *Il6*, (**F**) *Il1b*, (**G**) *Tnfa*, and (**H**) *Ifng* mRNAs by qPCR. Data in (**A**–**C**) are presented as mean ± SEM, while data in (**D**–**H**) are presented as geometric mean ± geometrical SD. Statistical differences compared with untreated Hjv^+/+^ or Hjv^−/−^ mice are indicated by a or b, respectively.

**Figure 4 ijms-24-01669-f004:**
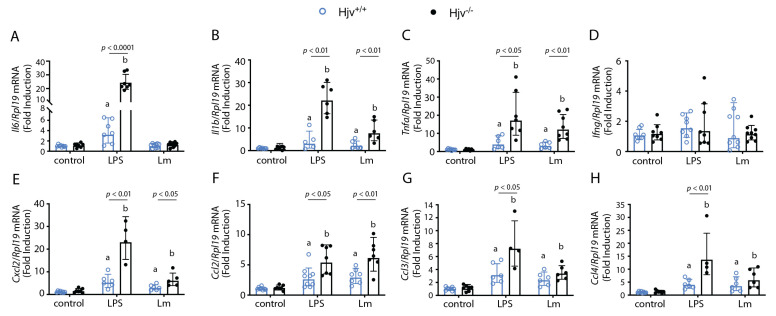
Cytokine expression in peritoneal lavage following acute infection with *L. major*. *Hjv^+/+^* and *Hjv*^−/−^ mice were injected intraperitoneally with either phosphate-buffered saline (control), the endotoxin LPS, or 10^8^
*L. major* stationary phase parasites (Lm). Then, 6 h post-infection, the peritoneum was lavaged, and peritoneal cells were collected. RNA was extracted, reverse-transcribed, and used for qPCR analysis of (**A**) *Il6*, (**B**) *Il1b*, (**C**) *Tnfa*, (**D**) *Ifng*, (**E**) *Cxcl2*, (**F**) *Ccl2*, (**G**) *Ccl3*, and (**H**) *Ccl4* mRNAs. Data are presented as geometric mean ± geometrical SD. Statistical differences compared with untreated *Hjv*^+/+^ or *Hjv*^−/−^ mice are indicated by a or b, respectively.

**Figure 5 ijms-24-01669-f005:**
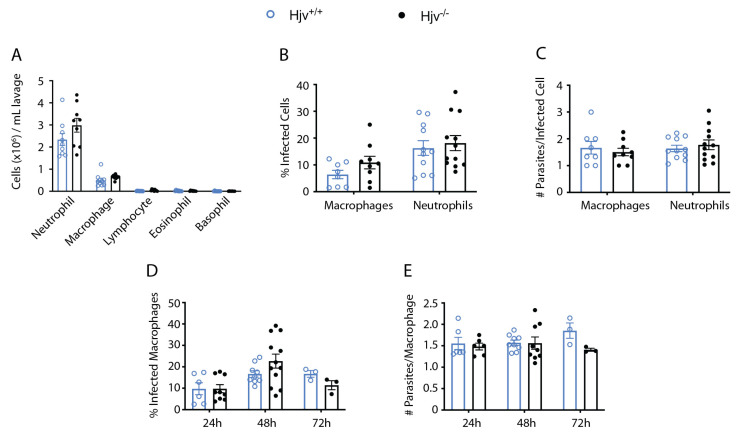
Analysis of intraperitoneal macrophages and neutrophils recovered post-infection. *Hjv^+/+^* and *Hjv*^−/−^ mice were injected intraperitoneally with 10^8^
*L. major* stationary phase parasites. Then, 6 h post-infection, the peritoneum was lavaged, peritoneal cells were counted, and 50 μL from the suspension was used for cytospin centrifugation. Cells were fixed onto slides and stained using Diff-Quik. Numbers of separated cell types are shown in (**A**). Percentage of infected macrophages and neutrophils (**B**), and number of parasites per infected cell (**C**) were assessed. (**D**,**E**) 10^5^ cells following lavage were plated in 4-well chambers. Cells were cultured over 24, 48, and 72 h before removal of media, drying, and staining with Diff-Quik. Percentage of infected cells (macrophages) (**D**), and number of parasites per infected cell (**E**) were assessed. Data are presented as mean ± SEM.

**Figure 6 ijms-24-01669-f006:**
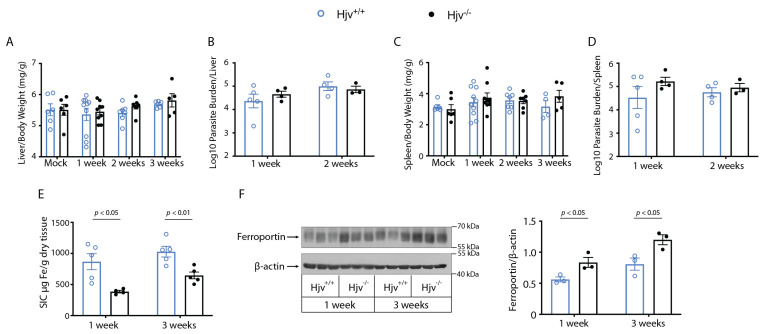
Severe genetic iron overload does not affect visceral leishmaniasis disease progression. *Hjv^+/+^* and *Hjv*^−/−^ mice were injected intraperitoneally with phosphate-buffered saline or 10^8^
*L. infantum* parasite. Mice were sacrificed 1, 2, or 3 weeks post-infection: (**A**) liver/body weight; (**B**) parasite burden in the liver; (**C**) spleen/body weight; (**D**) parasite burden in the spleen; (**E**) splenic iron content (SIC); (**F**) a representative Western blot of splenic ferroportin; data from n = 3 experiments were quantified by densitometry and are shown on the right. Data are presented as mean ± SEM.

## Data Availability

All data are contained within the manuscript and the Appendix A.

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
