# Peer review of "Genetic Iron Overload Hampers Development of Cutaneous Leishmaniasis in Mice"

_ijms, 2023, doi:10.3390/ijms24021669_

Round 1
Reviewer 1 Report
Charlebois et al in the manuscript entitled ‘Genetic Iron Overload Hampers Development of Cutaneous Leishmaniasis in Mouse Footpads’ presented an interesting observation in which they detected Hjv-/- mice had delayed L. major growth in hind footpads than wild ones; however they did not find any detectable difference for L. infantum infection between Hjv-/- and wild type mice. They further verified different immune-parameters but any specific reason for the observation could not be singled out. There are several concerns in terms of presentation of the manuscript that need to be addressed.
In Fig. 1A, Footpad thickness data were presented for 8 weeks but in legend it’s mentioned as 7 weeks. In Method section also in lines (317-318) it’s mentioned that Group of 5 mice were kept for measurement of footpads for 8 weeks. Please clarify.
Fig. B-C: Degree of infection in wild type mice remained same on 4 and 7 weeks suggesting either due to immunity or limitation of microenvironment the number of parasites may not increase whereas in Hjv mutant mice it’s catching up to and then become equivalent at 7 weeks. Whether the observation can be explained by this way? Is any data available for later than 7/8 weeks of infection that would probably clarify this issue better.
C57BL/6J mice were used for the current study that is reported to be less susceptible to Leishmania infections, whereas BALB/c mice display a susceptible phenotype (Restrepo CM et al, Gene expression patterns associated with Leishmania panamensis infection in macrophages from BALB/c and C57BL/6 mice PLoS Negl Trop Dis 15(2): e0009225. https://doi.org/10.1371/journal.pntd.
0009225) that could be a limitation of the study as any generalization of iron overload and similar observation may not be found in another mouse model. It should be included in the discussion.
In Abstract (lines 12-13; ‘Macrophages, Leishmania’s host cell, may divert iron traffic by reducing uptake or by increasing efflux of iron via the exporter ferroportin.’) and in Introduction (lines 37-38; ‘Macrophages are equipped with multiple mechanisms to combat this intracellular infection, including diverting iron flux to starve invaders [7, 8]); these sentences should be altered in context of published report of Leishmania-induced ferroportin degradation (Ben-Oathman et al, Plos Patho, 2014, Ref no, 49 of the current manuscript) and inhibition of translation (Das et al, Cell Microbiol, 2018, not included in reference list). This will better present the manuscript to general readers.
In similar context LN 144-145 “In fact, host macrophages are known to upregulate ferroportin expression in order to reduce their iron content in response to intracellular pathogens such as S. typhimurium [37] may be better presented with the Lishmania induced alteration of Ferroportin instead of S. typhimurium in regard to the current manuscript.
It would be also better to introduce a small background for general readers why in an overload condition cytokines-chemokines are analysed.
In Methods within the Serum biochemistry section serum ferritin was reported to be determined but apparently no data was shown in the manuscript (Line 364).
Reviewer 2 Report
The work "Genetic Iron Overload Hampers Development of Cutaneous Leishmaniasis in Mouse Footpads" is relevant and good, but it can be improved. I believe that the period between 1 and 7 weeks of infection, shown in Figure 1, especially at week 4, can be better explored, evaluating the expression and production of cytokines in the paw (site of infection). I believe it would bring better information about the process.
Title suggestion: Genetic Iron Overload Hampers Development of Cutaneous Leishmaniasis.
Introduction:
Line 36: I suggest putting only “macrophages” instead of “macrophages or neutrophils”, as the final destination is macrophages.
Line 52: I suggest using L. infantum instead of L. chagasi, including the cited reference addressing L. infantum.
Materials and Methods:
4.1 Animals – Lines 299 – 303: Animal research ethics committee approval number is missing.
Results:
1. The results shown in Figure 3H need to be better explained, as the infection by L. major only induced the expression of IFN-Æ” in the Hjv+/+ group. In the Hjv-/- group, IFN- Æ” expression is similar to the genotype-matched control (Line 151).
2. Another point: Figure 1A shows that Hjv-/- mice have a partial load control between weeks 1 to 7, however this same group has a lower expression of IFN- Æ” (Figure 3H). There is a statistical difference, it must be mentioned in the text.
3. I believe it would be interesting to evaluate the cytokine profile in paws infected with L. major, at week 4, where there is a difference in paw size and parasite load between Hjv+/+ and Hjv -/-. The expression and also the production of cytokines by ELISA could be done.
